# Religion and Spirituality as Relevant Dimensions in Psychiatric Patients—From Research to Practice

Samuel Pfeifer [ID]

Marburg Institute of Religion and Psychotherapy, 35039 Marburg, Germany; samuelpfeifer@gmail.com

**Abstract:** Background: Associations between psychiatric syndromes and religion/spirituality (R/S) are confounded by a diversity of descriptive instruments, the interpretation of statistical correlations, and the highly individual experience of illness. Method: This presentation focuses on three major syndromes in psychiatric patients: (a) delusions with religious content, (b) depressive conditions, and (c) anxiety disorders. Results: The content of delusions is marked by cultural factors, including religious concepts. There is empirical evidence that R/S may have a supportive role in patients with schizophrenia. Affective disorders show a more varied pattern of causality—a better outcome in about 60%, but in 10%, there seems to be a higher incidence in patients with a conservative, guilt-oriented, religious background. In anxiety disorders, a meta-analysis could not find a correlation between R/S and clinical syndromes. However, research into the emerging field of "spiritual struggles" has shown an interaction between subjective anxieties and religious conflicts, strongly influenced by the level of neuroticism beyond religious factors. Conclusions: The correlation of R/S and dysfunctional psychological experience may be summarized in three concepts (culture, conflict, and coping), modulated by the neurobiological basis of psychiatric disorders.

**Keywords:** religion; spirituality; delusions; schizophrenia; affective disorders; anxiety; neuroticism; spiritual struggles; coping

## 1. Introduction

Exploring the nature of major psychiatric disorders, with a special reference to religious/spiritual aspects, has its own challenges. Clinical work, even if it is research oriented, is going deeper than just applying "objective" measurement scales to a group of stratified study participants. Rather, clinical perspectives are qualitative, describing the subjective experience of illness in the individual person (Kleinman 1988), listening to both the burden and the suffering as well as the ways of trying to understand and to cope with the illness. Thus, clinical psychiatry has a deeply person-centered approach (Akhtar et al. 2020), which is not always matched by so-called objective research. Listening to individual stories, and deeply personal perspectives of suffering, and trying to existentially empathize with psychiatric patients cannot be pressed into the rigid molds of statistics. In contrast, contemporary research is somewhat dominated by figure-based research and, in many fields, by an emphasis of neurobiology over phenomenological, descriptive publications. The contemporary clinician, therefore, must look at quantitative research and meta-analyses, trying to interpret them in the context of person-centered psychiatry (Christodoulou et al. 2008).

Thus, approaching the relationship between psychiatric syndromes and religion/spirituality (R/S) requires the consideration of several preliminary aspects: the question of causality, the interpretation of statistical correlations, and the role of coping resources, to name but a few. The wording "relevant dimensions in psychiatric patients" of the title points to the subjective, existential aspects of the experience of the individual.

The meta-analyses of complex correlations in mental health pose their own challenges (Cuijpers 2016). There are concerns about the measurement of spirituality (Koenig 2008)—many studies do not sufficiently measure intrinsic or extrinsic religiosity and do not

differentiate between being spiritual and being religious, often only asking for membership in a church or any religious affiliation. This makes meta-analyses difficult.

Many studies, even if they try to measure as many factors as possible, end up with a scarcity of statistically robust conclusions. Thus, a broad study of 7403 participants in the United Kingdom (King et al. 2013) "confirms that religious people are less likely to use alcohol and recreational drugs but fails to confirm North American evidence that holding a religious understanding of life provides protection against mental disorders. It also concurs with other evidence from England that there is no clear relationship between religiosity and happiness" (p. 71).

Li et al. (2016), in a large sample of 48,984 nurses in the United States, found that "there is evidence that higher frequency of religious service attendance decreased the risk of incident depression and women with depression were less likely to subsequently attend services." Thus, more depression in people who do not attend church services does not yet "prove" that "people with less religiosity" have "more depression." Rather, there is a reverse causality (Maselko et al. 2012), concluding: "These findings suggest that women are more likely to stop attending religious services after onset of depression. Selection out of religious activities could be a significant contributor to previously observed inverse correlations between religious service attendance and psychopathology during adulthood." VanderWeele et al. (2016) reflect on causal inferences in longitudinal studies, concluding that there may be effects in both directions, making cross-sectional data ineffective for drawing inferences about causation.

In a re-examination of an earlier study on the effect of R/S on the risk of developing depression, Anderson et al. (2021) explored the validity of earlier positive tendencies during midlife in the same cohort of participants. Interestingly, the positive effect of religious commitment in younger years had now given way to a more complex picture, showing a variation "across adult development, with risk for depression associated with R/S at midlife potentially revealing a developmental process."

## 2. Hypotheses

The interaction of religion and psychiatry has repeatedly been examined in a comprehensive way. A broad overview of recent developments was provided by Koenig et al. (2020). The authors came to the following conclusion: "Overall, studies indicate that religious involvement often serves as a powerful resource for patients, one that can be integrated into psychiatric care. At times, however, religion may impede or complicate treatment."

This article focuses on three major areas of psychopathology, which all show a complex combination of biological, psychological, and social aspects. The following hypotheses will be examined in more detail:

(a) Religious delusions in psychotic disorders are based on the underlying neurobiological disorder, however, in their content, influenced by the cultural aspects of religion.

(b) Affective disorders are more complex. First, there may be a biological vulnerability; second, a depressive affect creates self-deprecation and feelings of guilt, which may be culturally identified with religious teachings; third, depression keeps patients from living their religious faith in the way they would like to; and finally, religious teachings or religious conflict may create a spiritual struggle and induce or worsen the depressive affect in the individual.

(c) Anxiety disorders show two major clinical impressions: (a) Clinical anxiety disorders, as described in diagnostic handbooks, are based on a biological vulnerability, combined with social triggers, and (b) subthreshold anxiety states are often influenced by the basic personality trait of neuroticism. Here, religion can function as a major trigger to internalize conflict and struggle, confirming models of "religious anxiety," which are now conceptualized in the term "spiritual struggles."

(d) Religion serves as a strong element of comfort and coping in mental distress. This has been demonstrated in patients with schizophrenia, affective disorders, and anxiety states.

From the literature, three major markers of the functions of religious teachings in their interactions with mental disorders can be suggested: culture, conflict, and coping.

## 3. Delusions with Religious Content

There seems to be no area of psychopathology that draws such public attention and morbid fascination as the field of religious delusions (RDs). The discrepancy between grandiose revelations and disorganized behavior, between holy words and unholy demeanor, between mystical experiences and offensive conduct causes pitiful rejection at best and religious unrest at worst.

Historical accounts of "religious insanity" are found in a two-volume 1200-page textbook by a German psychiatrist Karl W. Ideler (1848, 1850), who was medical director of the psychiatric department of the renowned Berlin Charité. He attributes "religious insanity" to ancient mystics living in the desert, the flagellants of the 11th century, possession epidemics in medieval monasteries, and radical religious movements during the Reformation, to name but a few examples he was discussing. Another German psychiatrist, von Kraft-Ebing (1879), describes "paranoia chronica (acuta) halluzinatoria religiosa" (p. 293) and talks about "theomania." However, in their fascination with the often bizarre and grotesque religious content of delusions, the authors mostly failed to adequately express the respect for healthy religion and to address the difference between functional and dysfunctional aspects of religion.

Although William James (1902) did not directly address the topic of religious delusions in his seminal work on the varieties of religious experience, he commented on religious mysticism to be only half of the great mystical stream, insanity being the other "diabolical" half of mysticism.

In a transcultural paper with philosophical underpinnings, Littlewood and Dein (2013) developed a theory on how the Christian religion could have influenced the development of schizophrenia, under the title "Did Christianity lead to schizophrenia? Psychosis, psychology and self reference." Referring to transcultural reports and interpretations over several centuries without objective clinical data, the paper seems rather speculative, equating Christianity with "Westernization."

Over the past 30 years, an increasing number of studies have been published that attempt to approach the subject of religious delusions in a more objective and scientific manner using phenomenology, anthropology, and cultural sociology to explore the nature of this multifaceted phenomenon.

Religious delusions have been described in all major cultures on all continents. However, prevalence varies widely by country and sociocultural context (ranging from 5% in China to 21% in Germany and as high as 44% in Malaysia), suggesting that different definitions of R/S and of religious delusion have been applied (Mohr and Pfeifer 2009). The highest prevalence was reported to be 70% of patients with schizophrenia in the Xhosa tribe of South Africa (Connell et al. 2015). A meta-analysis on the topic of finding meaning in religious delusion (Bhavsar and Bhugra 2008) concluded that religious rituals and family expectations play a significant role in the development and maintenance of delusions. The authors argue that there should be a reassessment of the importance of religious delusions in the light of new ethnographic and clinical evidence.

Given different definitions, the need arose to develop an algorithm to better conceptualize delusions and hallucinations with religious content. Siddle et al. (2002) describe the following factors necessary to diagnose a religious delusion: (a) Beliefs are held with absolute certainty, may be bizarre, and cannot be corrected by rational arguments or doubt; (b) additional clinical symptoms of a psychotic disorder; (c) religious content (God, Satan, prophecies, spirits, angels); (d) the ideas are not acceptable in the patient's subculture (peer group); and (e) lifestyle/goals indicate a psychotic episode rather than an enriching life experience.

Although delusional experiences seem to be caused by neurobiological mechanisms, the content of delusions is marked by cultural factors, including religious concepts. An

interesting catamnestic study compared patients with religious delusions in two German centers, namely the Charité in Berlin and the Psychiatric District Hospital in Regensburg, in the years between 1980 and 1985 at the time of two German states (Pfaff et al. 2008). While the Charité admitted patients from the largely secularized inhabitants of East Berlin, the population in the Regensburg catchment area in Bavaria was 80% Catholic. Religion was defined in general terms of transcendence, including a belief in supernatural forces, spirits, and superstition. The results were apparent: religious delusion occurred significantly less frequently in East Berlin than in Regensburg (11.6% vs. 28.6%, $p$ = 0.0046). The authors concluded that the occurrence of religious delusions in the context of schizophrenic disorders was essentially culture related. RD should therefore be classified as a secondary symptom of schizophrenia.

A Dutch study prospectively investigated the effects of religious delusions on the course of psychosis in an elderly population (Noort et al. 2022). In 137 patients with an average age of 76.3 years, a distinction was made between religious delusions and other delusional topics. There was a non-significant tendency toward a less favorable course in the presence of religious delusions. In patients with schizophrenia, religious delusions persisted more frequently than the most common delusions. No significant difference was found between patients with RD and patients without RD with regard to indicators of clinical complexity. Although psychopathology of religious delusions may be fascinating, it often obstructs the view of coping. Thus, the contribution of individual religiosity/spirituality to coping in people with schizophrenia has long been neglected. A research team in Geneva (Mohr et al. 2006) undertook to interview 115 outpatients with an underlying psychotic illness using a semi-structured interview on the role of religion in coping with their limitations. The results were surprising: For 71%, their faith provided hope, meaning, and significance to their lives, and 14% experienced religion as conflictual. In addition, the patients reported that faith helped them reduce the psychotic and baseline symptoms of their illness, while 10% experienced an increase, 28% reported improved social integration through religion, and 3% felt more isolated. Suicide risk decreased by 33%, while 10% reported being more suicidal because of religious struggles. The authors concluded that religion could be a vital resource for coping in patients with schizophrenia and could therefore play an important role in mental health care.

In summary, it can be said that the phenomenon of religious delusions is broad and extremely individual, primarily influenced by the environmental culture. The complexity of the contributing factors makes it difficult to find a uniform methodological strategy in different studies. The basic tenor of recent studies does not indicate a causality of religion for the occurrence of psychosis; rather, RDs are most likely to be understood as a culturally shaped secondary symptom.

## 4. Affective Disorders

A review article (Bonelli et al. 2012) examined 444 quantitative studies on the relationship between religion and depression, published between 1962 and 2011. The results showed that 60% reported less depression and there was faster remission of depression in those with more R/S or a reduction in depression severity in response to an R/S intervention. In contrast, only 6% reported more depression. Of the 178 most methodologically rigorous studies, 119 (67%) found an inverse relationship between R/S and depression, i.e., people with more pronounced religiosity show less depression. Religious beliefs and practices can help people cope better with stressful life circumstances, provide meaning and hope, and surround people with depression with a supportive community. However, for some populations or individuals, religious beliefs can increase feelings of guilt (Braam et al. 2000; Stompe et al. 2001) and lead to discouragement as people fail to live up to the high standards of their religious tradition (Hess 2014; Dein 2013).

A cross-cultural study by a Dutch team (Braam et al. 2010) compared religious coping strategies among four cultural groups using the R-COPE (Pargament 1998) and the SCL-90-R inventory: Dutch (N = 309), Moroccan (N = 180), Turkish (N = 202), and Surinamese

(N = 85) as a mother tongue. Although there were some correlational trends, causality could not be inferred. Across all cultural backgrounds, there was an association between depressive symptoms and negative religious beliefs: For example, "I wonder if God has abandoned me," punishment by God, anger at God, or doubt in the existence of God. The authors concluded that depression often expresses an inner emptiness regardless of the religious background.

What about the prospective effect of religion as a protective factor against depression? This question was investigated by Braam and Koenig (2019) in a systematic meta-analysis of 152 prospective studies. The results showed that 49% of the studies found a significant correlation between R/S and a better course of depression. In 41% of the studies, there was no significance, and in 10%, the results were negative or unclear. It has to be underlined that negative effects of religion have been observed, and there seems to be a higher incidence in patients with a conservative, guilt-oriented, religious background. R/S was significantly more likely to be protective in those with clinical symptomatology (d = −0.37). R/S was less likely to be protective in younger samples and in samples of patients with medical conditions. Studies with more extensive consideration of additional variables were significantly more likely to show an association with less depression. Geographic differences in results were not found.

Approaching the question of a positive effect of religion on mental health, i.e., reduction in the incidence of depression, Schnittker (2001) differentiated between "attendance at religious services, religious salience, and spiritual help seeking." In a sample of 2836 respondents, he found a U-shaped curve for depression and the degree of religious salience. Depression rates were higher in people without religious salience but also in subjects with a high degree of salience. A lower incidence of depression could be found in the middle of the curve, with a moderate degree of religious salience. In contrast, the degree of spiritual help seeking increased in a linear correlation with the intensity of depression. A similar U-shaped curve was reported by Schettino et al. (2011) when they examined religiosity and the treatment response to antidepressant medication. It was people with medium religious intensity who responded best, while a high degree of religiosity was rather associated with a higher persistence of depressive symptoms despite medication. This would support clinical observations that a high level of religiosity with narrow structures and restrictive moral demands creates additional stressors and conflicts, specifically for patients with depressive syndromes and melancholic personality traits, which would be detrimental to symptom improvement.

Considering the large number of studies over the past 50 years that were analyzed for the meta-analyses, this brief overview seems inadequate to address the question of how religiosity affects the development, course, and remission rates of depression. However, interested readers will find a wealth of further literature references in the aforementioned reviews. In summary, Braam (2009) describes the state of knowledge in an overview as follows:

- Religiousness relates to some degree to better mental health in the community and represents a source of adaptive coping in times of adversity (extensive evidence).
- The recovery rate from depression is substantially better for patients who attach intrinsic value to their religious faith and patients involved in a religious community (some evidence).
- During depressive episodes, negative feelings, such as discontent toward God or feeling abandoned by God, are highly prevalent (good evidence).
- Religious beliefs and practices are equally common among psychiatric inpatients, including those with depression; the frequency of prayer may be even higher irrespective of whether it leads to recovery from depression (some evidence).
- Patients with depression with a Christian background may be more likely to present with feelings of guilt (some evidence).
- Religion may have a small protective effect against suicidal thoughts and behaviors but should not be overestimated in the context of other risk factors) sociological evidence).

## 5. Anxiety Disorders

Anxiety is one of the most fundamental human emotions. In its warning function, it ensures survival in an unpredictable environment. However, when it becomes dysfunctional, it becomes a torment and overshadows life in many areas, leading to intense bodily discomfort, narrowing the psychosocial radius, and preventing people from unfolding their potential. In psychiatric diagnostics, "classical anxiety disorders" are being described (such as generalized anxiety disorder, panic disorder, or social phobia), as well as trauma- and stress-related disorders and the spectrum of obsessive-compulsive disorders (OCDs). There is much evidence for a familial clustering of anxiety disorders, with the symptomatology being shaped by a combination of genetic predisposition factors and individual psychosocial stress (van Sprang et al. 2022).

However, there is a second conceptualization of anxiety, which used to be addressed in the psychoanalytical theory of neurosis and in the psychodynamics of mental abnormalities. Today, they are reflected in the taxonomy of personality traits, especially in the concept of "neuroticism," which is one of the five scales of the widely accepted NEO-Five-Factor Inventory (Costa and McCrae 2008).

Let us start with clinical anxiety syndromes as defined by the DSM or ICD. There is a scarcity of studies on the correlation of religion and clinical anxiety disorders. A recent meta-analysis examined the influence of religious aspects and personal beliefs on psychological behavior, with a focus on anxiety disorders (Agorastos et al. 2014). The authors point to the "poor operationalization, incomparable data, and contradictory results" impeding the interpretations of the studies. They conclude that religion has been shown in many studies to be an important coping factor in the management of anxiety but that no robust correlation between clinical anxiety disorders and religiosity has emerged. They confirmed an earlier study by Shreve-Neiger and Edelstein (2004), who found only 17 studies that examined an empirical correlation between anxiety disorders and religiosity. A major flaw in the studies were conceptual and methodological weaknesses, not distinguishing between correlation and causality. They proposed a broader selection of study groups in terms of demographics and religious affiliation to produce more valid results. In the already cited British study by King et al. (2013), the authors could not establish a connection between anxiety disorders and R/S, neither in a positive nor in a negative sense, but they also commented that a religious or spiritual life is no guarantee for mental health.

Quite different is the research field of neuroticism and religion. If neuroticism is conceptualized as a vulnerability factor (Nordahl et al. 2019), it has an influence on a person's metacognitions (Wells 2009). In this context, conflictual or subjectively threatening metacognitions lead to a negative anxious affect, which is then experienced as pronounced stress. These maladaptive thinking styles are closely related to a basic personality constellation (trait anxiety), which are operationalized in the concept of neuroticism. It must be noted that although these unpleasant affects are often associated with anxiety, they do not reach the level of clinical anxiety disorders, i.e., they are "subthreshold" disorders (Witlox et al. 2021).

The conceptualization of anxiety in its relation to religion has undergone a major change over the years. Empirical research has developed a model for the influence of R/S on conflictual functioning, often associated with anxiety. Exline et al. (2014) have validated a "scale for religious and spiritual struggles," describing six core domains. In their individual struggles with religion, respondents named the following areas: (a) divine (negative emotion centered on beliefs about God or a perceived relationship with God), (b) demonic (concern that the devil or evil spirits are attacking an individual or causing negative events), (c) interpersonal (concern about negative experiences with religious people or institutions and interpersonal conflict around religious issues), (d) moral (wrestling with attempts to follow moral principles and worry or guilt about perceived offenses by the self, often associated with sexual issues), (e) doubt (feeling troubled by doubts or questions about one's R/S beliefs), and (f) ultimate meaning (concern about not perceiving deep meaning in one's life).

Subsequent research studied the occurrence of spiritual struggles in association with temperamental traits as measured by the NEO-FFI (Big Five). In a large study conducted via an online survey with 1047 adults and 3083 undergraduate students (Wilt et al. 2017), the following findings emerged: Subjects with high scores on agreeableness and conscientiousness presented low religious conflicts, whereas subjects with high scores on neuroticism reported a higher lifetime frequency of R/S struggles and a higher degree of current R/S struggles (about 26–36%), even higher than subjects with high religiosity (about 13–16%). Religiosity as a sociocultural factor influences the subjects, e.g., doubts about faith (which could be Christian, Islamic, or Jewish) or the assumption of demonic causality of psychological problems (Exline et al. 2021). Finally, religious conflict was found (not surprisingly) to have an impact on well-being (although significantly less than neuroticism). These empirical findings add to the growing body of research relating religious conflicts to individual personality differences, especially individuals with a tendency to anxiety (neuroticism).

Once again, however, a causality cannot necessarily be derived from this correlation. Rather, one could speak of a problematic constellation of personality and the religious/spiritual aspects of sociocultural context. Religion is experienced as oppressive, guilt inducing, and restrictive when there is a conflict with individual aspirations, drives, and desires that run counter to religious guidelines, rules, and control mechanism regulating conduct (Geertz 1973).

## 6. Conclusions

Does religion cause psychopathology? There is no easy answer from empirical research, as it is confounded by manifold methodological problems. One limitation of this overview is the fact that articles have been chosen from a clinical point of view to provide a perspective on the questions discussed, without following a structured comprehensive evaluation of publications. Nonetheless, care has been taken to include major meta-analytic papers that are based on such a structure. In none of the three major groups of mental disorders, the causality of R/S on the development of the problem could be demonstrated. Thus, we need to shift focus: From the literature, three major markers can be suggested for the relationship between R/S and mental health problems (culture, conflict, and coping), modulated by the neurobiological basis of psychiatric disorders. Religion and culture are closely intertwined, to the extent that it is culture that shapes the content of a condition and gives the impression of a "religious" disorder. Religion as a value-oriented, socio-cultural construct has considerable conflict potential: research shows that religious teachings or subcultural rules and customs may cause conflict, especially in individuals with a specific personality pattern of increased neuroticism. In contrast, protective elements are evident in the expression and course of depressive conditions. R/S may be experienced as an important subjective source of coping with the stress of mental disorder and its psychosocial effects.

**Funding:** This research received no external funding.

**Conflicts of Interest:** The author declares no conflict of interest.

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
