# Peer review of "Religion and Spirituality as Relevant Dimensions in Psychiatric Patients—From Research to Practice"

_religions, doi:10.3390/rel13090841_

Round 1
Reviewer 1 Report
- This article presents an overview of available quantitative research on religion/spirituality (R/S) and psychopathology, focussing on three major psychiatric syndromes. It finds three core concepts concerning R/S and psychopathology, and concludes that actual evidence is not conclusive with regard to causation of psychopathology by R/S. Strength of this article is its broad scope and overview, without losing an eye for detail.
- This article lacks one or two central questions / hypotheses, and a clear line of argumentation. It conclusions seem to pop up somewhat unexpectedly, although thorough reading reveals the used arguments.
More detailed:
Central problem of this article is the ambiguity with regard to its central question(s) or hypotheses. It seems to examine the central question if R/S has a causal role in inducing or reducing psychopathological states. Its summary of relevant literature meanwhile focusses on qualities of R/S in psychopathology and its form in all three major psychiatric syndromes of interest, and not on quantitative literature to (dis)approve a causal role. It then pops up 'three concepts: culture, conflict, and coping. This answers another question: what central concepts with regard to (influence and appearance of) R/S in psychopathology can be discerned? - Additional specific comments:
. Abstract, line 8: 'a higher incidence in [..] guilt oriented religious background' is not substantiated in the main text ánd does not fit the results of Braam&Koenig 2019 (it combines that percentage with Braam 2009, and applies qualitative differences between religious groups incorrectly to causal inferences on 'a higher incidence'
. Main text, line 309-10: These three major markers are not stepwise deducted from previous evaluated literature, but seem to pop up here.
. In the introduction of this article, a reference to Koenig 2020 (doi: 10.1192/bja.2019.81) would give a more actual and refined starting point, the rest of the article could be used to work up towards three (or more) major themes of R/S in psychopathology, given the fact that the more quantitative (meta)analysis/review on causality has been done already by Koenig. Conclusion of Koenig: 'Overall, studies indicate that religious involvement often serves as a powerful resource for patients, one that can be integrated into psychiatric care. At times, however, religion may impede or complicate treatment.'
Author Response
Thank you for your most helpful comments. I have tried to integrate them as far as possible in my ammendments oft he text.
- Hypotheses were formulated along the line oft he article, thus making the conclusions of «culture, conflict, and coping» less surprising in the end.
- Koenig et al (2020) was inserted and referred to.
- Braam & Koenig (2019) has been mentioned in the text (page 5/214 in the revised text) – a more explicit referral to negative influence of religion is made in the text.
Reviewer 2 Report
Congratulations for this very interesting paper
Author Response
Thank you
Reviewer 3 Report
This is an interesting and timely review.
However, before consideration for publication, the paper must be improved.
Frist, the method of paper selection for the review is not clear. Did the authors use some guidelines (e.g. PRISMA)? How did they selected the papers they reviewed? This is a major limitation that may limit the conclusions of the manuscript.
Second, the conclusions should take into consideration how potentially abnormal experiences (e.g., self-disorder, illusions, unusual beliefs) can be integrated in a religious context, and how it can contribute to the R/S - psychopathology link. For example, some basic experiences in people with religious conversion and psychosis are similar, yet, only a minority of converters develop psychosis.
Author Response
Thank you for your comments. Some remarks:
Unfortunately, as a clinician with a broad knowledge of the literature I have not applied a systematic and comprehensive search of the available literature. However, I have taken care to include high-level meta-analyses which allow the interested reader a much further reaching scope of the international literature. I have now mentioned this fact as a limitation of the article in my conclusions.
You suggested to go further into details of the question how potentially abnormal experiences (e.g., self-disorder, illusions, unusual beliefs) can be integrated in a religious context, and how it can contribute to the R/S - psychopathology link. Unfortunately this discussion would need an extended discussion which would go beyond the scope of this very limited paper.
Thus, I am sorry, I cannot go beyond the amendments I have made. Should that be an obstacle for publication, I would leave the decision to publish or not to the editor.
Round 2
Reviewer 1 Report
Thanks to the author(s), there is a more clear line in this article. It gives clear clinical hypotheses at the start. At the end it loosely summarizes and interprets the findings.
One adaption is needed to do right to the content of this review. The conclusion in the abstract, line 103 and line 347 that three major functions of religious teachings emerge in their interactions with mental disorders: Culture, Conflict, and Coping, is very loosely connected to the findings. In my opinion this conclusion could be no more than a suggestion, unless a beter founded argumentation is added. An addition like 'from literature three major markers can be suggested' or something like that, better fits the evidence and argumentation. I suggest such an adaptation.
Then this report is a clinically relevant insight in an multivalent and multidimensional factor as religion and spirituality (R/S) is, for selected clinical populations; with a suggestion for overarching markers/characteristics of R/S.
Author Response
Thank you for this helpful comment. Your suggestion has been fully implemented in this revision.
Reviewer 3 Report
The authors made some revisions in their paper, which resulted in significant improvements.
However, from the discussion of the connection between religion and psychotic phenomena, some critical work is missing, namely, the work of Simon Dein and the reflections on it regarding schizophrenia, religion, and religious conversion. The authors are encouraged to discuss it because religious conversion phenomena have a high clinical relevance to differentiate that from psychosis.
Author Response
Thank you for pointing out the work of Simon Dein. The article by Littlewood and Dein (2013) on schizophrenia and the influence of Christianity have now been mentioned in the paper.
We also looked at this article: Dein, S. (2018). Against the Stream: Religion and mental health – the case for the inclusion of religion and spirituality into psychiatric care. BJPsych Bulletin, 42(3), 127-129. doi:10.1192/bjb.2017.13
However, as it does not give more details on the question you addressed, it was not included in the revised version of the article.